# Preparation of Binary Thermal Silicone Grease and Its Application in Battery Thermal Management

**DOI:** 10.3390/ma13214763

**Published:** 2020-10-26

**Authors:** Ziqiang Liu, Juhua Huang, Ming Cao, Guiwen Jiang, Jin Hu, Qiang Chen

**Affiliations:** Department of Mechanical Engineering, Nanchang University, Nanchang 330031, China; 350928919006@email.ncu.edu.cn (Z.L.); caoming@ncu.edu.cn (M.C.); jgw_cailiao@163.com (G.J.); 400928918003@email.ncu.edu.cn (J.H.); 410914119004@email.ncu.edu.cn (Q.C.)

**Keywords:** battery thermal management, interface thermal resistance, thermal silicone grease, phase change material, battery heat dissipation

## Abstract

To improve the problems of large interface thermal resistance and low heat dissipation efficiency in battery thermal management (BTM), this paper uses methyl silicone oil as the matrix, AIN, copper powder (CP), and carbon fiber (CF) as thermally conductive fillers, and acetone and stearic acid as particle surface modification components. A variety of binary thermal silicone greases (TSGs) with different compositions were prepared. Different instruments were used to test the material properties of TSGs, and a better TSG was selected to coat the interface between battery and phase change material (PCM) for battery charging and discharging experiments. Through the analysis of experimental data, it was found that among the TSGs made of three mixed fillers (AIN/CP, AIN/CF, CP/CF), the three TSGs had good thermal stability, and their thermal degradation temperature both exceeded 300 °C. As the ratio of thermally conductive filler was gradually changed from 5:1 to 1:5, the TSG containing CP/CF had higher thermal conductivity and lower volume resistivity, while the TSG containing AIN/CF had the least damage due to interface wear. The acidification treatment of thermally conductive filler can improve the adsorption and compatibility of thermally conductive particles and silicone oil, and reduce the oil separation rate of TSGs. The prepared expanded graphite (EG)/paraffin wax (PW) composite phase change material (CPCM) has a relatively large latent heat of phase change, which can effectively control the temperature of the battery, but coating TSG between the battery and the CPCM can further enhance the heat dissipation effect of the battery.

## 1. Introduction

With the continuous development of the electrical, electronic, energy and other industries, the demand for interface heat dissipation has become more urgent, and various interface thermal conductivity materials have also attracted the attention of many researchers [1,2]. Mao et al. prepared a core–shell structure thermally conductive and insulating thermal interface material composed of Al@Al_2_O_3_ and epoxy resin [3]. Liu et al. found that Al_2_O_3_/olefin block copolymer (OBC)/paraffin wax (PW) composite material had lower thermal resistance as an interface thermally conductive material [4]. Roy et al. compared the thermal performance and practical problems of various thermal interface materials (traditional: greases, phase change materials, gels, and thermal pads; emerging: carbon nanotubes, graphene, low melt alloys, and metallic nanosprings) [5]. Among them, thermal silicone grease (TSG) is a widely used thermally conductive composite interface material, which can maintain a semi-solid state at 200 °C and above, and maintain chemical stability for a long time. According to personal needs, the addition of additives into TSG can also make it conductive, insulating or flame retardant [6,7]. At the same time, the TSGs have no specific material composition and ratio; Han et al. successfully prepared a ternary TSG containing AlN/Ga-based Liquid Metal/Silicone oil composite materials [8]; Yu et al. prepared TSG containing graphene nanosheets and reduced graphene oxide [9]; Kang et al. made nano TSG by mixing copper nanopowder with silicone oil, using copper nanopowder a thermally conductive material and silicone oil is as the matrix [10]; Shishkin et al. tested the thermal conductivity of zinc oxide-based TSG, and prepared a high-performance TSG based on aluminum nitride as a thermally conductive filler [11]; and Yu et al. prepared three kinds of CuO-based TSG with different structures using a special synthesis method [12].

Thermal conductivity is an important characteristic of TSG. By adding highly thermally conductive materials and selecting appropriate thermally conductive fillers, the thermal conduction channel can be constructed and the energy transmission efficiency of thermally conductive phonons can be improved. Du et al. observed that tetrapod-shaped ZnO (T-ZnO) and ZnO short-columnar (ZnO-SC) have a synergistic thermal conductivity after mixing, and the unique three-dimensional thermal network formed by the two thermally conductive fillers can reduce interface phonon scattering [13]; Chen et al. introduced functionalized carbon nanotubes (CNTs) into silicone grease to accompany metal oxide particles (micron Al_2_O_3_, submicron ZnO) to enhance the thermal contact conductivity of composite grease [14]; Yu et al. found that two-dimensional graphene can bridge alumina particles to form a more compact thermal network structure [15]; and He et al. used graphene flakes (GFs), hexagonal boron nitride (h-BN) and hydroxypropyl cellulose (HPC) as fillers to improve the thermal conductivity of TSG [16].

For objects with serious heating, the existence of TSG can solve the problems of low heat transfer efficiency and slow heat dissipation. Gou et al. prepared TSG containing multi-walled carbon nanotubes (MWCNT) and used it for heat transfer and cooling of chips [17]. Zhang et al. used thermal interface materials in photovoltaic-thermoelectric coupling devices to improve the use of solar energy [18]. Kusuma et al. prepared TSG containing sodium silicate and zinc oxide as fillers. They believed that the application of TSG to the heat dissipation of electronic components such as central processing units (CPU) has great advantages [19]. Chen et al. believed that TSG containing graphene and alumina (Al_2_O_3_) can solve the need for rapid heat dissipation of small electronic devices and components [20].

In BTM science, in order to improve the heat dissipation capacity of the battery interface, Lv et al. filled graphene oxide modified silica gel between the liquid-cooled BTM pipes to improve the temperature uniformity of the battery pack [21]. Zhang et al. prepared a high thermal conductivity EG/PW/SR (Silicone Rubber) PCM, and proposed that the PCM has potential application value in the field of heat transfer at the battery interface [22]. However, research on the thermal conductivity of the battery interface is still lacking. In this work, according to the characteristics of TSG, a variety of binary TSGs were prepared and pioneeringly applied to the PCM BTM, as shown in Figure 1. When the power battery is working, the appropriate ambient temperature is 20–45 °C [23,24,25]. If the power battery continues to work with high intensity, the temperature can easily exceed the appropriate temperature range, which causes thermal runaway in the battery, and can even lead to battery fire or explosion [26,27]. If the battery environment temperature is too low, the battery capacity and voltage will be severely depleted, and battery life will be shortened [28,29]. In battery thermal management for PCM temperature control, the PCM can use its huge latent heat due to phase change to absorb the heat emitted by the battery, and well control the surface temperature of the battery [30,31]. However, there are many small pits and impurities between the seemingly flat battery surface and the PCM, making the two objects unable to fit together, and there are gaps and great thermal resistance between the interfaces, which seriously affect the heat dissipation of the battery. The introduction of TSG can greatly improve the thermal conductivity of the interface and improve the efficiency of battery thermal management.

## 2. Experiment Methodology

### 2.1. Preparation of TSG

#### 2.1.1. Experimental Materials

AIN is in powder form with a particle size of 0.5 µm–5 µm; CP is in granular form with a particle size range of 1 µm–17 µm; CF is in the form of long strips with a length range of between 30 µm and 225 µm, purchased from Jiangxi Nanchang Pinghai Biological Development Co., Ltd. Stearic acid is in the form of tiny lumps, and was purchased from Shanghai Yien Chemical Technology Co., Ltd. together with AIN and CP. In addition, acetone was purchased from Xilong Science Company, and methyl silicone oil was purchased from Zhejiang Rongcheng Organosilicon Material Company.

Figure 2 shows the particle size distribution of CP and AIN. The average particle size of CP was about 8 µm, while the average particle size of AIN was about 2.5 µm. Figure 3 shows the scanning electron microscope (SEM) images of CP, AIN, and CF. From the figure, it can be seen that the CP has irregular shapes and different sizes, but has good dispersion. AIN particles are smaller than CP, but AIN is more prone to agglomeration, so it needs to be dried for a long time during the experiment. CF is obviously elongated, and its size is much larger than CP and AIN powder. As a thermally conductive filler, it is beneficial for forming a thermally conductive network.

#### 2.1.2. Preparation of TSGs

Figure 4 is a flow chart of the preparation of TSGs. First, an appropriate amount of acetone is added to the beaker and the beaker is put into an oil bath, the oil bath is heated to 50 °C. Stearic acid particles are added to the beaker and stirred. After the stearic acid melts, the thermally conductive filler is added to the beaker. Then, the mixed liquid with the thermally conductive filler is stirred for 20 min and then allowed to stand for 1.5 h, and the upper layer liquid is sucked with a dropper. The obtained precipitate is placed in a drying oven at 100 °C for 3 h. Then, the dried thermally conductive filler powder is taken out and placed in a beaker. The weighed methyl silicone oil is added, and stirring is continued for 2 h in an oil bath at 100 °C. The ratio of silicone oil to thermally conductive filler is 2:5. Then, the beaker is taken out and stirred at room temperature for 0.5 h to obtain acidified TSG. However, if the two thermally conductive fillers are directly mixed in proportion, and then methyl silicone oil is added for stirring, the TSG prepared by this method will not have been acidified.

CP, AIN, and CF are used as the thermally conductive fillers of TSGs. Two of them are selected in turn and mixed in a certain mass ratio. According to the preparation process in Figure 4, the thermally conductive filler is acidified to prepare binary TSGs with better performance. In the experiment, three kinds of TSGs containing AIN/CP, AIN/CF and CP/CF were prepared, and the fillers in each TSG were mixed according to the mass ratios of 5:1, 4:2, 3:3, 2:4 and 1:5, respectively. Figure 5 shows the physical pictures of six TSGs among the 15 kinds of TSG prepared. It can be seen from the figure that the color of the TSGs has changes significantly with the change of the composition and content of various fillers.

### 2.2. Characterization

To understand the performance of the prepared TSGs, several material characterizations were performed in this article. A scanning electron microscope (SEM, FEI Quanta200F, Waltham, MA, USA) was used to observe the microstructure of each component of the TSG at room temperature. The X-ray diffraction technique (XRD, D8 ADVANCE Bruker, Karlsruhe, Germany) was used to observe the crystal structure of the material; the sample was tested in a nitrogen-filled environment, and the test angle was 10–90°. Fourier transform infrared spectrometer (FTIR, Thermo Fisher Scientific, Waltham, MA, USA) was used to observe the chemical composition and changes of the material; the wavelength of the light used in the test was 2.5–25 µm, and the wavenumber range was 400–4000 cm^−1^. A thermogravimetric analyzer (TGA4000, PE, Waltham, MA, USA) was used to test the thermal stability of TSG; the test temperature range was 0–600 °C, and the heating rate was 10 °C/min. A differential scanning calorimeter (DSC8000, PE, Waltham, MA, USA) was used to test the latent heat of the phase transition process of PCMs; the temperature range of this test was 20–70 °C, and the heating rate was 5 °C/min. A volume resistivity tester (ATI-212, JDYQ, Beijing, China) was used to measure the electrical properties of the material. A friction testing machine (MFT-R4000, Lanzhou Huahui Instruments, Lanzhou, China) was used to test the friction and wear properties of TSGs; AIS15200 steel ball, and a steel block chassis with a surface roughness of 0.05 µm were used in the test.

### 2.3. Battery Charge and Discharge Experiment

The prepared TSG is coated onto the interface between the battery and the PCM, and the battery is charged and discharged. Figure 6 shows the battery charging and discharging system and battery module. In this system, the battery was purchased from Dongguan Xingfeng Lithium Battery Technology Company, and the battery parameters are shown in Table 1. The battery charging and discharging equipment was purchased from Shenzhen Hengyi Energy Technology Co., Ltd., and the model number was PT120300A. The high and low temperature test chamber was produced by Shanghai Honghe Laboratory Equipment Co., Ltd., and the model is WGD-0208. The temperature data acquisition equipment was purchased from Japan’s HIOKI Company, and the model number was MR8902. When conducting the experiment, the battery was first put in a temperature control box at 25 °C, and a thin layer of TSG was evenly coated onto both surfaces of the battery. At the same time, two thermocouples with the test temperature were placed on the battery surface, the prepared PW/EG CPCMs were attached to the two surfaces of the battery, and the positive and negative electrodes of the battery were connected to the battery charging and discharging system. After the system was connected, the battery charging and discharging experiment was carried out. By analyzing the temperature response data recorded by the thermocouple, the effect of TSG on the temperature control of the battery can be judged.

## 3. Results and Discussion

### 3.1. Characterization and Analysis of TSGs

#### 3.1.1. Volume Resistivity and Thermal Conductivity

The three mixed thermally conductive fillers (AIN/CP, AIN/CF, CP/CF) were made into acidified TSGs with different compositions, according to the mass ratios 5:1, 4:2, 3:3, 2:4, and 1:5. The volume resistivity and thermal conductivity of the prepared thermal grease were tested, and the results are shown in Figure 7. When the mass ratio of the thermally conductive filler was 5:1, due to the strong electrical conductivity of AIN, the volume resistivity of AIN/CP was the smallest. When the mass ratio of the thermally conductive filler was 5:1, the volume resistivity of TSG containing AIN/CP was the smallest due to the strong electrical conductivity of AIN. When the proportion of the thermally conductive filler in the silicone oil was gradually increased, the spacing between the thermally conductive particles decreased, and the electrical conductivity of the three binary TSGs became stronger. In particular, when the ratio of CP/CF was 4:2, the volume resistivity of TSG containing CP/CF dropped sharply. This situation can be explained by the “tunnel theory”. When the CF content reaches a certain value, a “point–line–point” current channel is formed between CP and CF. When the CF content further increases, the conductive network structure in the TSG system becomes more abundant. Figure 7b shows the relationship diagram of thermal conductivity with the proportion of thermally conductive filler. It can be found that among the three TSGs, the TSG containing CP/CF filler had higher thermal conductivity. When the thermally conductive filler ratio was 1:5, the thermal conductivity of TSG containing CP/CF filler reached 1.81 W·mK, which is 0.46 W·mK higher than when the ratio was 5:1.

#### 3.1.2. Friction and Wear Performance

The MFT-R4000 ball-disk contact friction tester was used to conduct friction tests on different components of TSGs. In the test, AIS15200 steel balls with a diameter of 5 mm and a hardness of 710 HV were used. The chassis is a steel block, and the surface roughness of the steel block was 0.05 µm. The average friction coefficient and average wear scar width of the binary TSGs obtained from the test are shown in Figure 8. It can be seen that the TSG containing AIN/CF had a small friction coefficient. This may be because the particles of AIN are small, and a large amount of AIN can have a certain self-repairing effect on the damaged parts of the material. In addition, CF and graphite have the same carbon composition, so CF also has a certain lubricating function. As the composition of AIN/CP changed, the friction coefficient of the TSG showed an increasing trend. This phenomenon is attributed to the large particle size and high hardness of CP particles. On the contrary, as the CP/CF composition changed, the friction coefficient of TSG gradually decreased. Figure 8b shows the wear scar width data of each component TSG. The graph is roughly the same as Figure 8a. TSG with a larger friction coefficient also had a larger wear scar width. The filler ratio was changed from 5:1 to 1:5, and the wear scar width of the TSG containing AIN/CP doubled, while the width of the wear scar of the TSG containing CP/CF was reduced by 45.8%.

#### 3.1.3. Thermal Stability Analysis

Figure 9 carried out TGA tests on three binary TSGs (the mass ratio of thermally conductive filler is 1:5) and methyl silicone oil. The mass ratio of thermally conductive filler to silicone oil in TSG was 5:2, and the temperature range of the TGA test was 0 °C to 700 °C. As the test temperature was increased, at 315 °C, the mass fractions of the three different components of TSG began to change. At about 350 °C, the degradation process of the three TSG ended, and the remaining mass fraction remained at a constant value of 71%. Among the TSGs, the TSG containing CP/CF filler had the fastest degradation rate, which may be due to its having better thermal conductivity than the other two TSGs. Comparing the TGA curves of silicone oil and TSGs, it can be seen that the degradation rate of pure methyl silicone oil was relatively slow, and its mass fraction tended to 0 after the degradation was completed. Silicone oil degraded at 315 °C, while AIN, CP, CF were not thermally degraded in the range of 0 °C to 700 °C. However, in battery thermal management, the battery ambient temperature is generally below 100 °C, so the prepared TSGs are able to meet the requirements of thermal stability.

#### 3.1.4. Crystalline Characterization

The TSG containing CP/CF (1:5) with good thermal conductivity was taken, and XRD tests were performed on the acidified and unacidified TSG, respectively. The results are shown in Figure 10. It can be seen from Figure 10a that the XRD characteristic peak of CP is relatively gentler than that of CF, while the characteristic peak of CF is larger and its crystallinity is better. The XRD characteristic peak of methyl silicone oil approaches a straight line. This is because methyl silicone oil is in a liquid state, so no obvious characteristic peak can be seen during the test. The XRD characteristic peak of the TSG in the figure is similar to that of CF. It can be explained that the mixture of CF, CP, and methyl silicone oil is only a pure physical mixture, and the XRD characteristic peak of the TSG is just the superposition of the XRD characteristic peak of the components that make up it. Figure 10b is the XRD characteristic peak of the acidified TSG and its components. It can be seen from this figure that the degree of crystallization of stearic acid is better and the peak is obvious. The XRD characteristic peak of the acidified TSG is not formed by superimposing the XRD characteristic peaks of each component. This is because acetone and stearic acid will be completely volatilized during the heating and drying process during the preparation of TSG. In addition, the height of the XRD characteristic peak of the acidified TSG is inconsistent with the height of the XRD characteristic peak of the unacidified TSG. This is because the unacidified thermally conductive filler powder is prone to agglomeration, and the particles have poor compatibility with methyl silicone oil. Acidification treatment of inorganic thermally conductive fillers can improve particle adsorption, wetting and dispersion properties, and reduce the interface thermal resistance of polymer composites, but the chemical properties of thermally conductive fillers did not change significantly; therefore, the shape of the XRD characteristic peak of TSGs in Figure 10a,b is not exactly the same.

#### 3.1.5. Chemical Structure

FTIR was used to analyze the structure and chemical bond of the acidified TSG containing CP/CF thermally conductive filler and its components. The test range of the one-dimensional infrared absorption spectrum was 400–4000 cm^−1^, and the test results are shown in Figure 11. It can be seen from the figure that methyl silicone oil has four obvious infrared absorption peaks. The infrared absorption peaks in the range of 700–800 cm^−1^ are mainly attributable to the Si-C stretching vibration band of methyl silicone oil and the rocking vibration band of CH_3_. The absorption peak in the range of 1000–1100 cm^−1^ is mainly attributed to the stretching vibration band of Si-O of methyl silicone oil. The infrared absorption peak in the range of 1200–1300 cm^−1^ is mainly attributed to the CH_3_ deformation vibration band of methyl silicone oil. The absorption peak in the range of 2800–3000 cm^−1^ is mainly attributed to the CH_3_ stretching vibration band of methyl silicone oil. The spectrum of methyl silicone oil is similar to that of TSG, but the FTIR curve of TSG does not have the characteristics of acetone and stearic acid curves. This phenomenon also verifies the speculation in Figure 10 that stearic acid and acetone were completely volatilized during the experiment, the CP/CF hybrid thermally conductive filler and methyl silicone oil are simply mixed physically.

#### 3.1.6. The Oil Separation Time of TSG

The same mass of acidified and unacidified TSGs was weighed on paper, the paper was put into the drying box, and the temperature of the drying box was set to 70 °C. The oil separation of the TSGs was observed through the glass window of the drying box. In this case, when there was obvious precipitation of silicone oil, the oil separation time of each component of the TSG was recorded. The data is shown in Table 2, and Figure 12 shows the actual oil separation diagram of the TSGs after heating. It can be seen from Table 2 that among the six TSGs, the higher the content of CF, the longer the oil separation time. This may be due to the larger volume of CF under the same mass, so it has better adsorption and compatibility with silicone oil. In addition, the acidified TSG containing CP/CF (1:5) did not precipitate silicone oil until 5 h later, while the acidified TSG containing AIN/CP (1:5) precipitated silicone oil after 1.2 h. Comparing the oil separation time of the acidified and unacidified TSGs, it can be found that the acidified TSGs had a longer oil separation time, which shows that the acidified thermally conductive filler can effectively improve the compatibility between the thermally conductive filler and the silicone oil.

### 3.2. Phase Change Characteristics of Composite Phase Change Materials (CPCMs)

The CPCM absorbs the heat generated by the power battery by sticking to the surface of the battery, so as to achieve the purpose of controlling the temperature of the power battery. The CPCM used in the battery charge and discharge experiment was made of PW and EG, and the manufacturing method adopted was the melt blending method. The PW in the CPCM was refined paraffin, and the specification of the EG was 50 mesh. The DSC test results of PW and PW/EG CPCM are shown in Figure 13a. PW starts to absorb a lot of heat and melts at about 42 °C, while its temperature rises slowly at this time. Both the endothermic and exothermic processes of PW have oscillating waves and two peaks, which may be due to the impure composition and inconsistent melting point of the selected PW. In addition, the amplitude of CPCM was significantly smaller than that of PW. This was because the content of PW in CPCM decreased, and its phase transition enthalpy became proportionally smaller. Comparing the endothermic and exothermic temperature values of the DSC curve, it can be found that both PW and CPCM have a certain degree of supercooling. Figure 13b shows the cyclic DSC test curve of PW/EG CPCM. It can be seen that the phase transition enthalpy, melting point, freezing point and supercooling degree of CPCM did not change significantly during repeated testing, indicating that the prepared CPCM had good stability and circulation.

### 3.3. The Influence of TSG on the Battery Charging and Discharging Process

Two pieces of 8-mm-thick CPCMs were pasted on both surfaces of the battery. The width and height of the CPCMs were the same as those of the battery. A layer of TSG was evenly applied between the battery and the CPCM interface, and an acidified TSG containing CP/CF (1:5) with good comprehensive performance was selected for the experiment. The battery was connected to the charging and discharging system as shown in Figure 6, and experiments were conducted and the battery temperature data was recorded. The temperature of the battery surface is the average of the two thermocouples. Since the maximum discharge rate of this battery was 2C, the discharge experiments were carried out at discharge rates of 1C and 2C, respectively, and the temperature response data of the battery with and without thermally conductive silicone grease were recorded, as shown in Figure 14. It can be seen from the figure that when the battery was discharged at 1C, the surface temperature of the battery showed a linear increasing trend. The battery module with TSG added between the CPCM and the battery interface was 0.52 °C lower than the final temperature without addition, but the final temperature of neither module reached the PW phase transition temperature. When performing 2C discharge, the battery surface temperature rose much faster than for 1C, and the temperature of the battery module with TSG and the battery module without TSG slowed down significantly at about 42 °C, indicating that the huge latent heat of the phase change of CPCM plays a role in the temperature control of the battery. In addition, the temperature of the battery module with TSG at the end of discharge was 42.17 °C, which was 1.48 °C lower than that of the battery module without TSG. This shows that the CPCM temperature control method can effectively control the surface temperature of the battery in the optimal temperature range (20–45 °C), and adding TSG to the interface between the battery and CPCM can improve the thermal management ability of the battery to a certain extent.

## 4. Conclusions

In this paper, a variety of binary component TSGs with different mass ratios were prepared using AIN, CP, and CF as thermally conductive fillers, and a series of material characterizations were made on the prepared TSGs. During the experiment, TSG was uniformly coated on the interface between the battery and the CPCM and the battery charge and discharge experiment was performed. Finally, the effect of TSG on PCM battery thermal management was evaluated according to the temperature response. According to the SEM morphology and particle size properties of AIN, CP and CF, binary composition TSGs of AIN/CP, AIN/CF and CP/CF were designed and prepared. When the ratio of the fillers changed from 5:1 to 1:5, the volume resistivity of the three kinds of TSGs decreased significantly. The TSG with CP/CF composition decreased the fastest, while the change trend of thermal conductivity was opposite to that of volume resistivity. The thermal conductivity of TSG containing CP/CF increased by 0.46 W·mK. The friction experiment of the three binary TSGs found that as the ratio of thermally conductive filler changed from 5:1 to 1:5, the friction coefficient of TSG containing AIN/CP gradually increased, that of TSG containing CP/CF gradually decreased, the friction coefficient of TSG containing AIN/CF did not change much, and the curve of the wear scar width of the three binary TSGs was similar to the curve of the friction coefficient. It can be seen from the TGA curve that TSGs are thermally degraded at about 315 °C, and they have good thermal stability. It can be seen from the oil separation experiment of the TSGs that the silicone oil adsorption and physical stability of the TSG containing CF were stronger, and the higher the CF content, the longer the oil separation time of TSG. In the battery module, the prepared PW/EG CPCM had great latent heat of phase change. The performance of the CPCM was stable after multiple DSC cycle tests, and its enthalpy and supercooling degree did not change. According to the temperature response curve of the battery module, the PCM temperature control can effectively control the battery surface temperature in the optimal temperature range (20–45 °C), and the battery discharge end temperature of the battery cooling module with TSG is lower than that of the battery module without TSG, which indicates that adding TSG to the interface of battery and CPCM can improve the battery thermal management ability to a certain extent.

## Figures and Tables

**Figure 1 materials-13-04763-f001:**
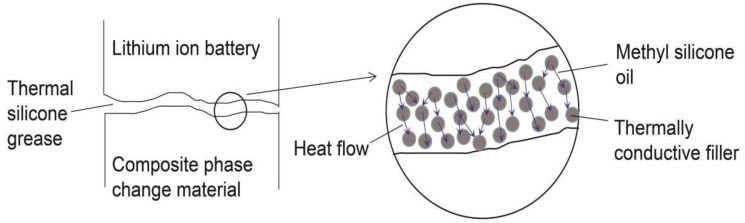
Schematic diagram of heat conduction at the interface of battery/TSG/PCM.

**Figure 2 materials-13-04763-f002:**
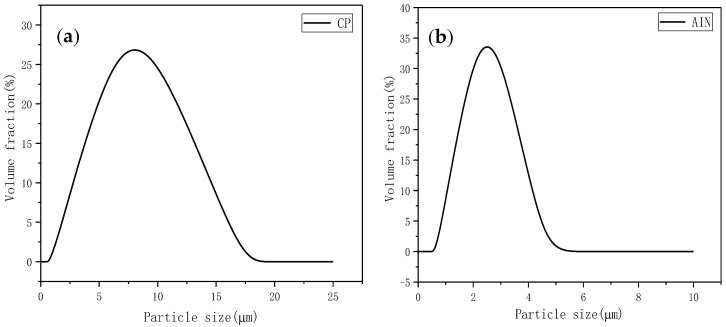
Particle size distribution diagram. (**a**) CP. (**b**) AIN.

**Figure 3 materials-13-04763-f003:**
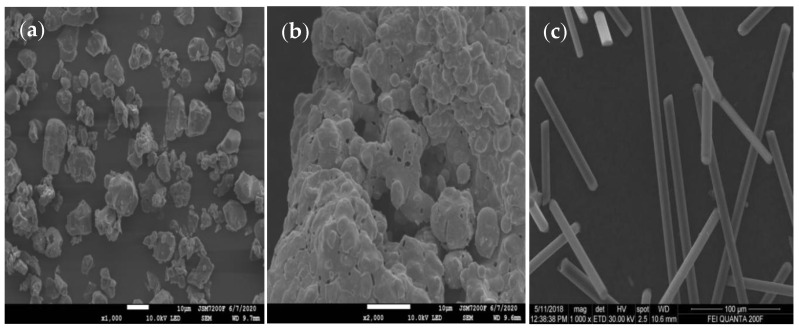
SEM images. (**a**) CP. (**b**) AIN. (**c**) CF.

**Figure 4 materials-13-04763-f004:**
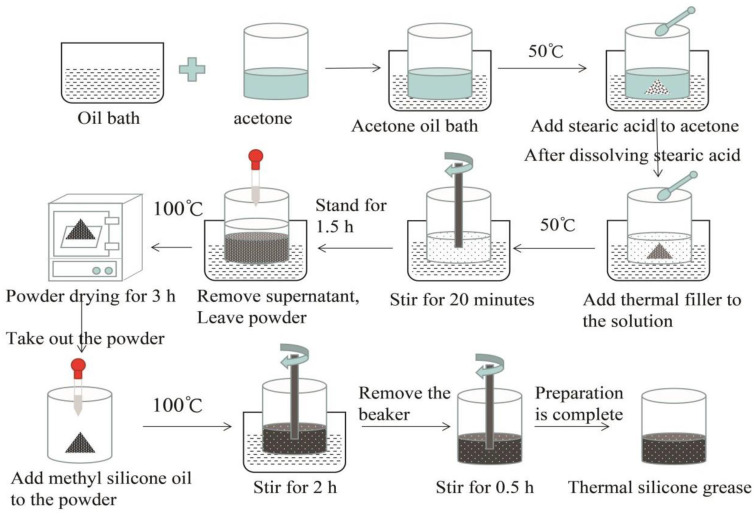
Flow chart of preparation of TSG.

**Figure 5 materials-13-04763-f005:**
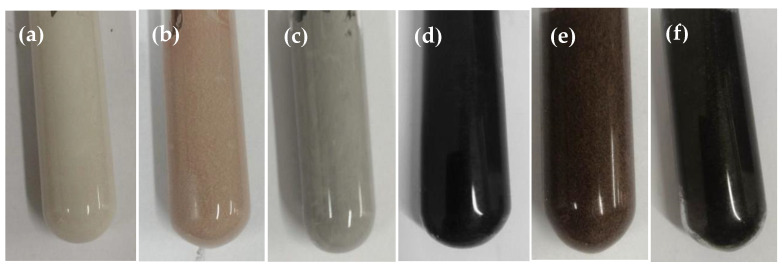
TSG with different types of fillers. (**a**) AIN:CP = 5:1. (**b**) AIN:CP = 1:5. (**c**) AIN:CF = 5:1. (**d**) AIN:CF = 1:5. (**e**) CP:CF = 5:1. (**f**) CP:CF = 1:5.

**Figure 6 materials-13-04763-f006:**
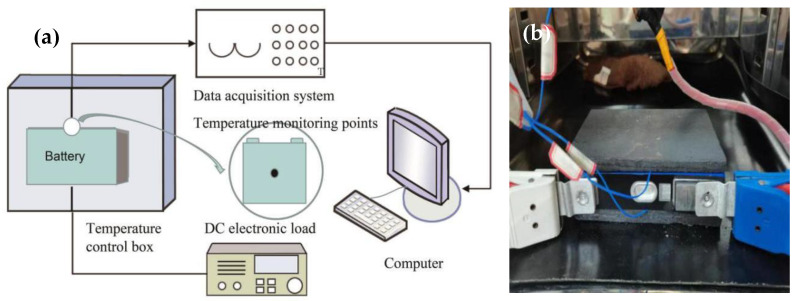
Battery charge and discharge experiment. (**a**) Schematic diagram of battery charge and discharge system. (**b**) Battery module.

**Figure 7 materials-13-04763-f007:**
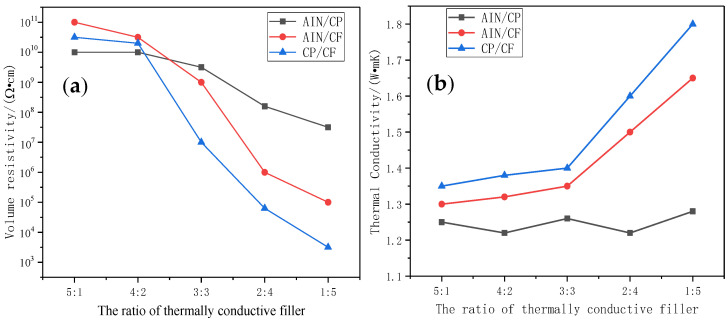
Volume resistivity and thermal conductivity of TSGs with different compositions. (**a**) Volume resistivity. (**b**) Thermal conductivity.

**Figure 8 materials-13-04763-f008:**
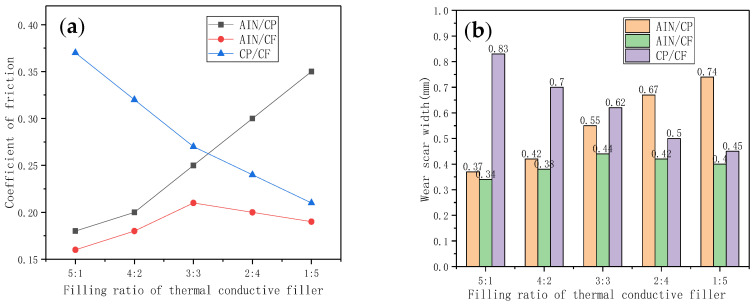
Friction and wear performance of binary TSGs with different thermal conductivity filler ratios. (**a**) Friction coefficient. (**b**) Width of wear scar.

**Figure 9 materials-13-04763-f009:**
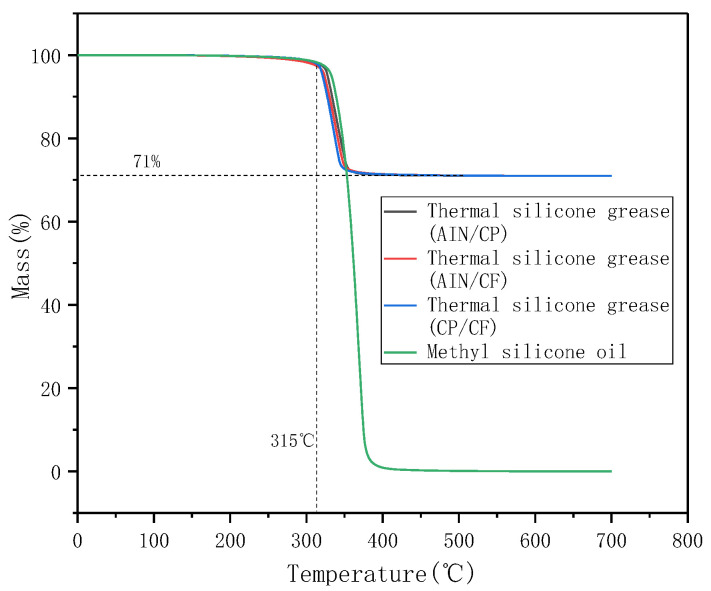
TGA curve of TSGs and silicone oil.

**Figure 10 materials-13-04763-f010:**
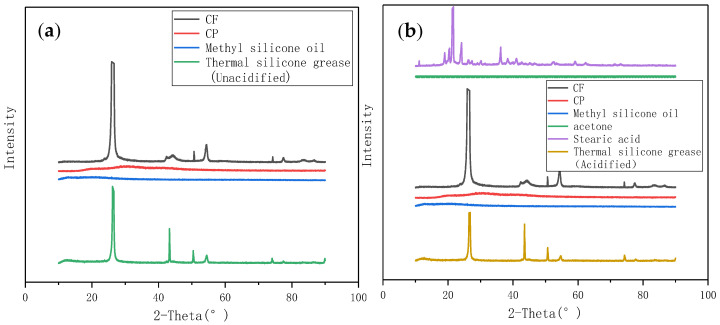
XRD pattern. (**a**) XRD characteristic peak of unacidified TSG and its components. (**b**) XRD characteristic peak of acidified TSG and its components.

**Figure 11 materials-13-04763-f011:**
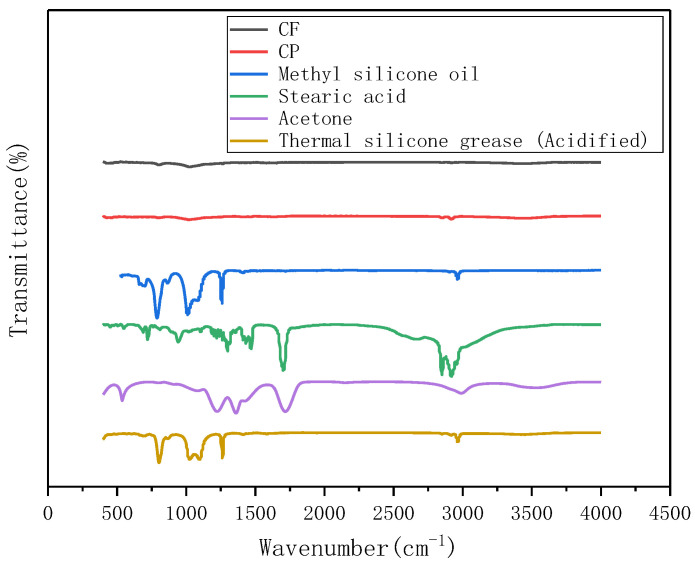
FTIR spectrum of acidified TSG containing CP/CF thermally conductive filler and its components.

**Figure 12 materials-13-04763-f012:**
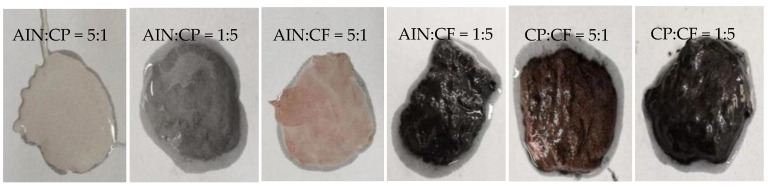
Physical image of oil separation after heating TSG.

**Figure 13 materials-13-04763-f013:**
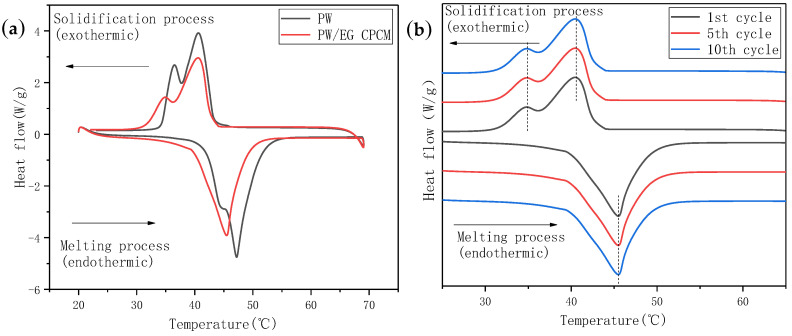
DSC spectrum. (**a**) DSC curves of PW and CPCM. (**b**) Cyclic DSC test curve of PW/EG CPCM.

**Figure 14 materials-13-04763-f014:**
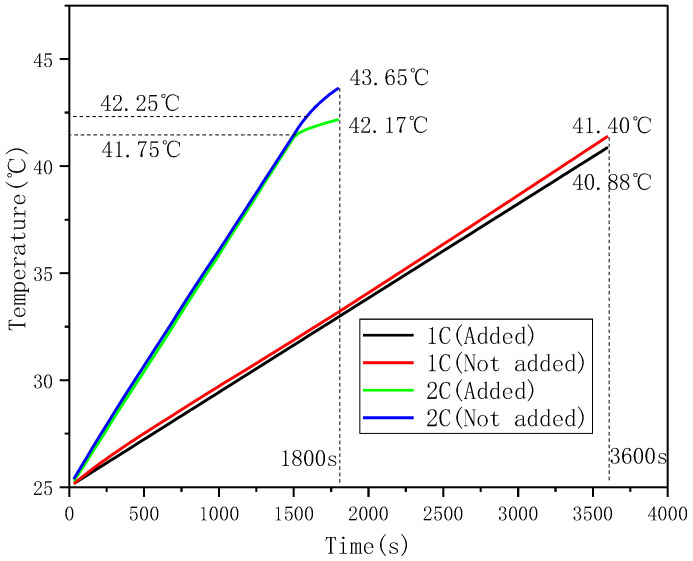
Temperature response curves of battery modules with and without TSG under 1C and 2C discharge conditions.

**Table 1 materials-13-04763-t001:** Battery parameters.

Thick, Wide, High (mm)	Internal Resistance (mΩ)	Capacity (Ah)	Weight (kg)	Charge and Discharge Cut-Off Voltage (V)	Nominal Voltage (V)	Maximum Charging Current (C)	Maximum Discharge Current (C)
27, 148, 130	0.6	65	1.2	2.7–4.2	3.7	1	3

**Table 2 materials-13-04763-t002:** Oil separation time of different components of thermal grease.

Material Composition	AIN/CP (5:1)	AIN/CP (1:5)	AIN/CF (5:1)	AIN/CF (1:5)	CP/CF (5:1)	CP/CF (1:5)
Oil separation time (acidified)	1.6 h	1.2 h	2.5 h	5.2 h	2 h	5 h
Oil separation time (unacidified)	1.5 h	1 h	2.5 h	4.8 h	1.8 h	4.5 h

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
