# Peer review of "Preparation of Binary Thermal Silicone Grease and Its Application in Battery Thermal Management"

_materials, 2020, doi:10.3390/ma13214763_

Round 1

Author Response

Dear reviewer, thank you very much for your comment on my manuscript.

Below is my response to your review comments:

  1. Abstract and Conclusions: Could the Authors condense the text with emphasis on the obtained results.

According to your suggestion, I have made some changes to the abstract and conclusion of the manuscript. For details, see Annex 1 (Annex 1 is the revised manuscript) and Annex 2 (Annex 2 is the revised manuscript with revision mark). However, there are still many material characterization results descriptions in the abstract and conclusions, because the preparation and characterization of thermal silicone grease is also the main content of the article and title.

  1. Introduction: In my opinion maybe little too long. Please make it more focused and related to the objective of your work. Add more up-to-date references

I have made some changes to the introduction of the manuscript and added more references closely related to the subject of the manuscript. For details, see Annex 1 and Annex 2.

  1. In the Materials and Methods section – 2.2. (p.5), please add more information, especially about he instruments used and conditions for the measurements.

I have made a supplementary explanation on the conditions of use of the instrument in section 2.2. For details, see Annex 1 and Annex 2.

  1. Figures and Tables: Please increase the Figures quality. Is it possible to move some of the Figures in the SI section to have main text more compact?

In the revised manuscript, I have made minor adjustments to the dimensions and specifications of figure 2, figure 3, figure 5, Table 1, figure 12, and figure 13, and improved the clarity and quality of figure 1, figure 4 and figure6. See Annex 1 and 2 for details.

  1. Please do English check and typo check through the whole text.

I have changed, replaced or deleted some English sentences and typos in the manuscript. The red and blue fonts in Annex 2 are the modification marks.

Note:

Since only one Annex can be selected, I put Annex 1 and Annex 2 in one document (Annex 1, 2).

Annex 1 is the revised manuscript;

Annex 2 is the revised manuscript with revision marks. In the text, red font is added or replaced content, blue font is deleted content.

Reviewer 2 Report

The manuscript deals with some systematic testing of composite materials with the purpose of evaluating their efficiency as battery thermal management materials.

The topic is quite well introduced to readers and the experiments are  accurately described.

The major limitation is represented by final results about the temperature rise difference with and without the TSG. In case of slower discharge (1C) the difference is 0.48 °C, which widen slightly to about 1,5 °C for faster discharge.

Is this meaningful for practical application? If yes, Authors should give some indications about standards and goals of commercial or special  applications. Without such a benchmarking, the outcome looks quite poor.

Other comments:

  • please use chemical formula for aluminum nitride “AlN” throughout the text. No need to define it as AIN;
  • about particle size distributions presented in Fig. 2, “most of” is a too generic expression. Pease use statistical quantities, such as “mode, mean or average”;
  • in section 2.1.2 authors mention the preparation of 6 mixtures with ratios 1:5 or 5:1 of fillers. However, in the rest of the manuscript several characterizations of samples with different ratios are reported. Please check and remove this apparent inconsitency, for instance by mentioning the preparartion of all composites in section 2.1.2;
  • features in XRD profiles are conventionally referred to as: peaks having some height or intensity. Please avoid using expressions like "curve" "waves" or "amplitude";
  • line 240-241:"The XRD curve of the acidified TSG is shorter than the XRD curve of the unacidified TSG". The meaning of this statement is obscure. Please reformulate or remove it;

Author Response

Dear reviewer, thank you very much for your comment on my manuscript.

Below is my response to your review comments:

At the end of discharging, although the battery temperature drops little after adding TSG, in actual use, this action is of great significance to the safety of the battery. There are three main reasons:

  1. As the battery discharge rate increases, the effect of TSG on temperature control will become more and more obvious.
  2. As the battery pack is prone to heat accumulation effect during use, when the temperature of each battery in the battery pack can drop slightly, the safety factor of the entire battery pack will be greatly improved.
  3. In traditional phase change material battery thermal management, the influence of interface heat conduction on temperature is rarely considered, but the addition of TSG can play a certain role in battery temperature control, while TSG does not occupy the volume of the battery module. If TSG is not used, to make the battery have the same cooling effect as adding TSG, more phase change materials need to be added, which will take up a large volume and weight of the battery module.

other:

  1. According to your suggestion, I have deleted the content of "defining aluminum nitride as AIN" in the article, and expressed it in the most simplified form of "AIN". For details, see Annex 1 (Annex 1 is the revised manuscript) and Annex 2 (Annex 2 is the revised manuscript with revision marks).
  2. I have revised the inappropriate description of particle size in the text. See Annex 1 and Annex 2 for details.
  3. I have described the types of TSGs we prepared in section 2.1.2, and Figure 5 only shows the physical images of 6 of the 15 TSGs we prepared, which guarantees the consistency of the proportion of TSGs described in the full text.
  4. I have made an accurate modification to the description of the XRD diagram in 3.1.4, replacing the expressions of "wave", "curve" and "amplitude" with "characteristic peak".

5."The XRD curve of the acidified TSG is shorter than the XRD curve of the unacidified TSG". I have deleted this vague expression in the manuscript, and made amendments and supplementary explanations there. See Annex 1 and 2 for details

Note:

Since only one Annex can be selected, I put Annex 1 and Annex 2 in one document (Annex 1, 2).

Annex 1 is the revised manuscript;

Annex 2 is the revised manuscript with revision marks. In the text, red font is added or replaced content, blue font is deleted content.

Reviewer 3 Report

I suggest to accept the manuscript after minor revision

1) The authors should add SEM image and EDS Mapping of the composite before and after thermal tests

2) What happen if Aluminium nitride is replaced by boron nitride?

3) Is the grease efficient for all of kind battery chemistries (e.g. NMC battery)?

4) I would recommend to add XRD patterns after thermal analysis

Author Response

Dear reviewer, thank you very much for your comment on my manuscript.

Below is my response to your review comments:

  • The authors should add SEM image and EDS Mapping of the composite before and after thermal tests

In this paper, through the particle size distribution, SEM test, volume resistivity, thermal conductivity, friction and wear performance, TGA test, XRD test, FTIR, silicone oil leakage rate, DSC and battery discharge temperature response test, TSGs have been characterized from various angles, and the effectiveness of TSGs used in battery thermal management has been proved, so EDS test of TSGs is not considered. In addition, we need to do SEM and EDS test in the unified material test center of the school. Because the test center is busy, it usually takes us half a month to get the test results, so it is difficult to complete all the material testing within the limited 10 day modification period. I hope you can understand. thank you!

  • What happen if Aluminium nitride is replaced by boron nitride?

Your idea is very good. Both BN and AIN are suitable as thermal conductive fillers for thermal silicone grease. However, because the thermal conductivity of aluminum nitride is usually higher than that of boron nitride, and the melting and boiling point of aluminum nitride is higher than that of boron nitride, so In the preparation of thermal silicone grease, aluminum nitride is used more frequently than boron nitride. In addition, when boron nitride is used as a filler, the heat transfer efficiency is lower than that of aluminum nitride.

  • Is the grease efficient for all of kind battery chemistries (e.g. NMC battery)?

Yes, thermal silicone grease is not only suitable for NMC batteries, but also for all batteries with solid-state casings. As long as there is a gap between the battery interfaces, the thermal silicone grease can be filled between the interfaces to improve the heat transfer and heat dissipation capabilities of the battery module.

  • I would recommend to add XRD patterns after thermal analysis

In the original manuscript, I tested the thermal stability of TSGs containing three mixed fillers of AIN/CP, AIN/CF, and CP/CF in section 3.1.3. However, in section 3.1.1 and section 3.1.6, I found that the TSG containing CP/CF had strong thermal conductivity and low oil leakage rate, so I only chose the TSG containing CP/CF with better comprehensive performance for battery discharge experiment. Therefore, in section 3.1.4, I only did XRD test on acidified and unacidified TSGs containing CP/CF, and did not do XRD test on all prepared TSGs.

In addition, I have made minor improvements to some sentences in the manuscript. For details, see Annex 1 and Annex 2.

Note:

Since only one Annex can be selected, I put Annex 1 and Annex 2 in one document (Annex 1, 2).

Annex 1 is the revised manuscript;

Annex 2 is the revised manuscript with revision marks. In the text, red font is added or replaced content, blue font is deleted content.

Round 2

Reviewer 2 Report

The authors did address all issues raised about the first version of the manuscript. It has reached now the minimum quality level for publication, although it still remain somehow unprecise and naive.